# Spatial Correlation of Air Pollution and Its Causes in Northeast China

**DOI:** 10.3390/ijerph182010619

**Published:** 2021-10-11

**Authors:** Mingze Du, Weijiang Liu, Yizhe Hao

**Affiliations:** 1Business School, Jilin University, Changchun 130012, China; dennisdu520@163.com (M.D.); haoyz20@mails.jlu.edu.cn (Y.H.); 2Center for Quantitative Economics, Jilin University, Changchun 130012, China; 3Northeast Revitalization and Development Research Institute, Jilin University, Changchun 130012, China

**Keywords:** northeast China, air pollution, spatial association

## Abstract

To understand the status of air pollution in northeastern China, we explore the structure of air pollution transmission networks and propose targeted policy recommendations. Using air pollution data from 35 cities in northeastern China for a total of 879 periods from 6 January 2015 to 3 June 2017, this paper used social network analysis (SNA) to construct a spatial association network of air pollution in the region, and analyzed the spatial association of air pollution among cities and its causes in an attempt to reveal the transmission path of air pollution in the region. The results show that inter-city air pollution in northeast China forms a complex and stable correlation network with obvious seasonal differences of “high in winter and low in summer”. Different cities in the region play the roles of “spillover”, “intermediary” and “receiver” of air pollution in the network. Small respirable particulate (PM2.5) pollution constitutes a significant component of air pollution in northeast China, which spreads from Liaoning province to Heilongjiang province via Jilin province. Therefore, regional joint pollution prevention and control measures should be adopted to combat the air pollution problem, and different treatment measures should be developed for different city “roles” in the pollution network, in order to fundamentally solve the air pollution problem in the region.

## 1. Introduction

Air quality is relevant to everyone’s life, and increased concentrations of air pollution significantly reduce the health of the population. According to the World Health Organization (WHO), air pollution kills nearly seven million people worldwide each year, and directly contributes to increased mortality from stroke, heart disease, lung cancer and acute respiratory infections [1]. With the increase in air pollution in recent years, both civilians, media and government agencies have become highly concerned about this issue. The Chinese government implemented an air pollution prevention and control program from 2013 to 2017, and although some public health results have been achieved [2], heavy air pollution has remained during this period, especially in the northeastern part of China. Due to its unique geographical location and heavy industrial structure, Northeast China is plagued by air pollution every year. Understanding the causes, dispersion and transmission processes of air pollution in the northeastern region has become a key issue in combating air pollution in the northeastern region.

The air pollution problem has caused serious economic and health hazards to the Chinese people [3,4], and some scholars have estimated that the health loss from air pollution in Beijing alone is equivalent to 0.03% of the gross local economic output [5]. Therefore, the study of air pollution is very important, and scholars have studied the causes [6,7,8], spatial and temporal distribution characteristics [9,10,11], and influencing factors [12,13] of air pollution, respectively. The findings show that air pollution has significant spatial correlation characteristics [14,15], and air pollution in a certain region is influenced not only by certain factors within the region, but also by certain factors external to the zone [16,17]. Studies have shown that when the industrial structure of a region [18,19] is dominated by the industry, industrial pollution can lead to serious air pollution problems [20,21]. Moreover, due to the pollution paradise hypothesis, companies in pollution-intensive industries tend to be established in areas with relatively low environmental standards, which bring foreign investment [19,22,23], and also cause air pollution problems. In addition, construction pollution [24,25] and motor vehicle emissions [13,26] are socioeconomic factors that contribute to air pollution. Due to the characteristics of air pollution, such as large scale and wide spread, geographical conditions, such as temperature, air humidity and airflow also determine the characteristics of air pollution [8,27,28,29].

However, most of the current relevant studies focus on the spatial effects of air pollution and only investigate the spatial spillover relationships of air pollution in neighboring regions, lacking the exploration of air pollution network relationships. From the experience of air pollution phenomena, such as air events in China, air pollution often spreads over several provinces [30], and its spatial spillover relationships are not only limited to adjacent provinces, but often appear to spread across provinces [31]. The air pollution network relationship refers to the process of air pollution propagation between cities in a certain region constitutes a dynamic association network with cities as nodes, forming a spatial association network of regional air pollution, and thus it is important to explore the association characteristics and driving mechanisms of this association network and other aspects. Some researchers have used social network analysis to analyze the spatial correlation of industrial NOx emissions in the Beijing-Tianjin-Hebei region and surrounding areas in China, and found that there is a stable spatial correlation network of pollution between cities. Urban industrial pollution emissions are significantly correlated between cities within 300 km of each other [32]. Some scholars also established a provincial pollution spatial network in China and analyzed its structural characteristics. The results of this study help to illustrate the current status of pollution control in China and provide support for the SNA approach to analyze the theory of pollution association networks [33]. In terms of air pollution, some scholars have explored the spatial correlation effects of air pollution in the Chengdu-Chongqing urban agglomeration in China and found that the spatial correlation network of air pollution in the Chengdu-Chongqing urban agglomeration has obvious regional distribution characteristics. However, the geographical scope of this study is relatively small and does not cover the spatial extent of air pollution [34]. Some scholars have also used social network analysis to study spatial association networks in areas such as carbon emissions. They found that greenhouse gases are also spatially correlated, and that carbon reduction measures can be targeted through the role played by different regions in the correlation network [35,36]. In the existing literature, studies on spatial association networks formed by air pollution have not been analyzed for the northeastern region of China. Moreover, even fewer studies that mention this region tend to analyze the provincial level of China’s pollution association networks by incorporating the provincial level, lacking relatively microscopic studies at the level of prefecture-level cities, and most of the relevant studies use annual data as the unit of measurement, thereby lacking detail. In addition, there is a gap in research on the role of air pollution transmission and prevention among the three provinces in Northeast China. The region has suffered from serious air pollution for a long time due to the presence of inadequate production technology and industry as the mainstay [37], as well as coal-burning for heating in winter [38]. The study of the spatial association network of air pollution in the region can help to understand the characteristics of air pollution in the region.

The current Air Pollution Prevention and Control Law only specifies that the local government is responsible for the ambient air quality in its own administrative region, without stipulating its responsibility for regional air pollution. In the face of the regional air pollution network that has been formed, the current “individual” model of urban air pollution prevention and control is no longer effective in solving the current pollution problem. In the ecological and environmental protection management into the officials assessment and evaluation system, in the face of the air pollution circulation characteristics, should be combined with regional air pollution network characteristics, “large regional, cross-regional” joint management as the scope of air pollution management.

Therefore, this paper analyzes the spatial association and causes of air pollution in northeastern China based on social network analysis method, and studies the network structure characteristics of the air pollution association network, the urban spatial clustering characteristics of air pollution and tries to explore the air pollution transmission paths in the region. This is an attemp to provide a policy basis for regional joint prevention and control and collaborative air pollution control by the Chinese authorities.

## 2. Model Construction and Theory Introduction

Spatial association is a comprehensive and systematic study of the structural characteristics and evolutionary mechanisms of social networks based on social network analysis methods. It abstracts the real complex system into a network by studying the relationship between individuals and analyzes the structural characteristics of the network resulting from the interaction between network nodes and the impact of the network system behavior on individual nodes. Most of the existing literature has used gravity models [39,40] or Granger causality test models in the Vector Autoregression (VAR) framework [34,41,42] to portray the interactions between network nodes. Given the gravity model is a measure constructed based on geographical or economic distance, which cannot reflect the dynamic interactions of this study on air pollution among nodes, the VAR model is used in this paper to analyze the spatial interactions among nodes. The VAR Granger causality is used to determine whether there is a dynamic association between nodes and to construct a dynamic causal interaction model of air pollution.

### 2.1. Dynamic Causal Interaction Model

Numerous studies have proven that as inter-city cooperation in economy, energy and transportation grows, cities become more closely connected to each other and eventually form clusters of closely linked cities [43]. Such urban agglomerations constitute a spatially linked network of inter-city factor flows. Due to the trans-regional transmission characteristics of air pollution and the combined effects of climate and socio-economic development, air pollution is no longer confined to one place, but can be transmitted between multiple places, even over long distances. In other words, air pollution is not an isolated pollution phenomenon in a separate area, a certain range of air pollution can affect each other, with a certain degree of correlation. Under the influence of climate factors and socio-economic factors, air pollution in a certain range can cause or aggravate the degree of air pollution in other ranges. This makes the air pollution boundary expand continuously, and the phenomenon of air interaction circulation between cities in the air pollution ridden area forms an inter-city air pollution spatial correlation network with cities as nodes. This dynamic correlation of interactions makes it possible to explore the propagation characteristics of air pollution from a time-series perspective.

The air pollution in one region may cause or aggravate air pollution in another region, that region has a certain ability to reflect the air pollution in other regions in advance, and the “predictive” ability of this early response can be explored by Granger causality test in time series. The Granger causality test can reveal whether the historical information of one time series has predictive power for the current value of another time series.

This study constructs a Granger causality test in a vector autoregressive model (VAR) framework to investigate the dynamic causal association between air pollution in two regions. Assuming that there are two regions x,y where air pollution occurs, the following VAR model is constructed to investigate the dynamic causal interaction between air pollution in the two regions, in order to test whether there is a Granger causality relationship between air pollution in the two regions, and the VAR model is as follows:(1)$$X_{t}=\alpha_{1}+\overset{m}{\underset{i=1}{\sum }}\beta_{1i}X_{t-i}+\overset{n}{\underset{i=1}{\sum }}\gamma_{1i}Y_{t-i}+\varepsilon_{1t}$$
(2)$$Y_{t}=\alpha_{2}+\overset{p}{\underset{i=1}{\sum }}\beta_{2i}X_{t-i}+\overset{q}{\underset{i=1}{\sum }}\gamma_{2i}Y_{t-i}+\varepsilon_{2t}$$
where $\left[X_{t}\right]$ and $\left[Y_{t}\right]$ are the air pollution time series of the two regions, *α*, *β*, *γ* are the parameters to be estimated, and *ε* is the residual term. While, *m*, *n*, *p*, and *q* are the lagged orders of the autoregressive terms. In the VAR model framework, the Granger causality between the variables is tested by a joint significance test of the coefficients of the autoregressive terms.

Since the series required for Granger causality test is a smooth series, this study tested the smoothness of the series before conducting the Granger causality test in the VAR framework, and differenced the non-smooth series. To maintain robustness, the lag order is chosen as 1–10 periods as the lag order to perform Granger causality test on the air pollution time series of 35 cities in northeast China, with 10% as the significance test. If both series $\left[X_{t}\right]$ and $\left[Y_{t}\right]$ pass the significance test, there is a causal relationship between the two series, i.e., X↔Y. Based on the above principle, the air pollution transmission network between regions can be constructed.

### 2.2. Social Network Analysis (SNA) Method

Social Network Analysis (SNA) is a new method for studying relational networks, mainly using graph theory tools and modeling techniques to reflect intergroup interactions and group structure, and is an interdisciplinary analysis method with “relational data” as the object. Its application field is expanding from sociology to economics and management, and it has become a new research model for the study of interactions. In this study, the association network formed by the dynamic association of air pollution in several cities in the region can be constructed by using social network analysis (SNA), using SNA tools to characterize the network structure of the spatial association of air pollution, and using QAP regression analysis in SNA to investigate the causes of the dynamic association of urban air pollution from the perspective of pollutants.

#### 2.2.1. Spatial Association Network Characteristics Indicators

(1)Holistic spatial association network characteristics: If the city within the air pollution region is considered as a node, all nodes, relationships, and connections within the region constitute a complex regional air pollution association network. In relation to the characteristics of the association network, there are indicators, such as network density, network efficiency, network relatedness, average distance, and network cohesion index based on the average distance, which are analyzed in terms of the closeness of inter-node connections, connection efficiency, robustness and network connectivity of the association network as a whole, respectively.(2)Local spatial correlation network node characteristics: In the air pollution correlation network, the local node characteristics are generally portrayed by the centrality of the node, centrality indicates the degree of a node in the network in the center of the network, the larger its value, representing the stronger the degree of centrality, the closer the degree of connection with other nodes intermediacy and control ability is better.

#### 2.2.2. QAP Regression Analysis

QAP regression analysis, the quadratic assignment procedure in the social network analysis method, is a method for studying the relationship between relational data. Non-parametric statistical methods are used to test the coefficients of the relational data based on repeated sampling and permutation of matrix data. The variables in this study are the relational data of air pollution transmission between cities within the region. The conventional statistical test method is not effective in handling the relationship data, and the high correlation between the relationship data and the spatial effect of the inclusion of geographic information data also makes the parameter estimation by the conventional method have problems such as “multicollinearity”, which leads to large errors in the parameter estimation and makes the significance test. The significance test is useless.

QAP regression analysis was performed by studying the regression relationship between multiple matrices and one matrix and testing the significance level of the coefficient of determination R^2^. The correlation coefficients of the matrices are given by comparing the corresponding values of the individual cells between the matrices and performing non-parametric statistical tests on the coefficients. The first step is to perform a conventional multiple regression analysis on the vector elements corresponding to the independent and dependent matrices, and the second step is to randomly permute the rows and columns of the dependent matrix and recalculate the regression, saving the calculated coefficient values and R^2^, repeating the process several times (usually 5000 times) and estimating the standard error of the statistic.

## 3. Empirical Analysis

### 3.1. Data Source

In this paper, a total of 879 periods of air pollution data from 6 January 2015 to 3 June 2017 in 35 cities in Northeast China were selected, in which the air quality index (AQI) was selected as a comprehensive indicator to measure the degree of air pollution [44,45,46], and six pollutants, such as PM_2.5_, PM_10_, SO_2_, CO, NO_2_ and O_3_ were influencing factors. Data sources from AQI China online monitoring platform https://www.aqistudy.cn (accessed on 12 February 2021) Missing values are replaced by the average of the two years before and after for processing [47,48].

### 3.2. Exploratory Analysis of Air Pollution in Northeast China

Firstly, the half-yearly averages of AQI indices from January 2015 to June 2017 were calculated for each of the 35 cities in Northeast China according to province and approximate geographical order. Then, a three-dimensional spatial surface map of the AQI index distribution in Northeast China was drawn, in terms of region, time period and AQI index (Figure 1).

The three-dimensional spatial surface plot shows that the AQI index varies between different regions at the same time, and also at different time periods in the same place. It is obvious from the figure that the prefecture-level cities attributed to Liaoning province have the worst air quality, followed by Jilin province, and the prefecture-level cities in Heilongjiang province have the best. This shows that air pollution in Northeast China varies not only in the temporal dimension, but also in the spatial dimension. The air pollution in Northeast China generally shows a decreasing and then increasing trend during the observation period, and the air pollution in some cities is significantly higher than that in other cities, indicating that the air conditions in different cities are different in the regional air pollution network. There are three different roles of “spillover”, “intermediary” and “receiver” in the spatially linked network. Most of the cities have obvious seasonal effects, showing the characteristic of “high in winter and low in summer”.

### 3.3. Inter-City Spatial Correlation of Air Pollution and Its Network Structure Characteristics in Northeast China

In this section, we perform Granger causality tests on air pollution among cities in Northeast China under the framework of VAR model, and construct a regional air pollution association network by dynamically correlating the causal relationships. We then investigate the characteristics of the constructed association network. Using Ucinet software, a dynamic correlation network model of 35 cities in Northeast China was constructed using cities as nodes and inter-city air pollution transmission relationships as connecting lines (Figure 2). It can be seen that the regional air pollution has formed a complex transmission network pattern among the cities in the region.

#### 3.3.1. Analysis of the Overall Characteristics of Air Pollution Spatial Association Network

In this paper, the overall characteristics of the air pollution spatial association network in Northeast China are analyzed from the perspectives of network density, network efficiency, network relevance, average distance and agglomeration index, etc. The basic characteristics of the overall air pollution spatial association network in Northeast China are detailed in Table 1.

##### Network Density Analysis

Network density is an indicator of how closely connected the nodes in a region are in a network. The higher the number of regional network associations, the higher the network density. Network density can be expressed as the ratio of the actual number of associations in the regional association network to the maximum number of associations the network can have.

In terms of the network density of AQI, the overall AQI network density in Northeast China is 0.769, indicating that there is a tight spatial correlation situation of air pollution within the region. The air pollution network shows characteristics such as multi-threaded, spanning large distances and dense transmission lines. It can be seen that a stable air pollution transmission network exists between cities in the region (Figure 2).

In terms of network density of sub-pollutants, the largest network density of urban air pollutants in Northeast China is PM_2.5_ with a value of 0.826, indicating that among all air pollutants, PM_2.5_ has the highest degree of association within the Northeast region, with more intensive conduction lines and degree of conduction, followed by SO_2_ and PM_10_. The network density of all pollutants exceeds 0.62, i.e., all pollutants are closely associated among cities in the region, which shows that the transmission paths are more stable in the region, both in terms of air pollution as a whole and individual pollutants. The network density of PM_2.5_ and SO_2_ is the highest among all pollutants, so the policies should focus on these two pollutants when formulating regional pollution prevention and control policies.

##### Network Efficiency Analysis

Network efficiency refers to the extent to which there are redundant connections in the network as a whole, given that the number of components in the associated network is determined. Therefore, when analyzing the overall network characteristics, network efficiency is used to portray the stability of the network. The lower the network efficiency, the more redundant connections exist in the network and the more stable the network.

In terms of the network efficiency of AQI, the network efficiency of the urban air association network in northeast China is 0.682, which indicates that more than 31% of the links are redundant associations, i.e., there is a large amount of superposition transmission in the air pollution association network in the region. The low network efficiency indicates the stability of the air pollution correlation network in the region, which also intuitively indicates the need to implement joint prevention and control measures for air pollution prevention and control at the regional level.

From the perspective of network efficiency of sub-pollutants, the network efficiencies of all pollutants are between 0.719 and 0.825, among which the network efficiency of PM_2.5_ is the lowest at 0.719, indicating that the association network of PM_2.5_ has high stability and more redundant association relationships and stronger pollution transmission in the spatial association network of pollution in the region compared to other pollutants. This is followed by pollutants such as PM_10_ and NO_2_ with network efficiencies of 0.752 and 0.757. The network efficiency perspective of the sub-pollutants also shows that the pollutants do not have high network efficiency and have high association network stability. Therefore, the pollution management model with individual cities as a unit does not cope well with the pollution problem in the region, and a regional joint prevention policy should be adopted to combat air pollution. Therefore, it should focus on PM_2.5_, PM_10_, NO_2_ and other components to manage.

##### Network Relevance Analysis

The network relatedness represents the accessibility of the associated network and is a measure of the connectivity of the entire associated network from the perspective of the network structure. Its value is between (0, 1) and the higher the value, the higher the degree of connectivity. In both AQI and sub-pollutant perspectives, the overall correlation magnitude of the air pollution transmission network in Northeast China is 1, which indicates that the degree of correlation of the air pollution transmission network in the region is high, and the air pollutants are closely connected in the whole regional transmission network, and the connectivity of the network is well established.

##### “Small World” Characteristics

The “small-world” feature refers to the identification of the connectivity of the network and whether it produces a “small-world effect” when examining the accessibility of the entire associated network. The “small-world effect” refers to the possibility of a small group of nodes when a part of the network is closely connected to each other. In the social network analysis method, “distance” is usually used to measure the small-world characteristics of the network, and this section selects “average distance” as the indicator to reflect the small-world characteristics of the air pollution network in the Northeast region. The cohesiveness index based on “distance” is measured to reflect the cohesiveness of the overall network.

According to the results in Table 1, the average distance of AQI in the northeast region is 1.232, which indicates that in this region, AQI needs only 1.232 intermediate cities on average to complete the connection between cities. Among the sub-pollutants, PM_2.5_ has the smallest average distance of 1.162, followed by PM_10_ with an average distance of 1.241, i.e., fine particulate matter is the most characteristic of the small world among all pollutants, requiring only 1.232 and 1.241 intermediate cities on average to establish a link between any two urban nodes in the region. The average distance of other pollutants is also between (1.2–1.4), which indicates that the spatial association network of air pollution in the region has obvious small-world characteristics. The cohesion index based on “distance” is 0.917 for PM_2.5_, followed by PM_10_ and CO, and the cohesion index of all pollutants is between (0.82–0.92), which proves that the spatial association network has strong cohesion and small-world characteristics. This small-world feature in the spatial association network makes the ability of air pollution linkage and influence between cities in the region significantly enhanced. It is inevitable to start from the perspective of joint prevention and control in a large region when formulating air pollution prevention and control measures.

#### 3.3.2. Analysis of Individual Characteristics of Air Pollution Spatial Association Network

The characteristics of an individual node in a spatial association network are usually measured by the centrality of the node in the network. Common measures of centrality are point centrality and intermediate centrality. Point centrality is divided into point degree centrality and point in and point out degrees. The point-in degree is the number of accepted associations, and the point-out degree is the number of overflow associations. In the association network, if the point-out degree is greater than the point-in degree, it is an overflow association. Conversely, if the point-in degree is greater than the point-out degree, it is an acceptance association, and if they are equal, the two relationships are equivalent. Intermediate centrality is the shortest number of times a node acts as an “intermediary” between two other nodes. The higher the number of times a node acts as an “intermediary”, the greater its intermediate centrality. In the following, the point degree centrality and intermediate centrality metrics are calculated for each city node in Northeast China, and the centrality of this regional association network is analyzed. The individual characteristics of the spatial association network of air pollution in Northeast China are detailed in Table 2.

In terms of point degree centrality, the top ranked cities in Northeast China are Harbin, Yingkou, Siping, Changchun, Shenyang, Liaoyuan and Suihua. The air pollution correlations between these cities and other cities are spillover relationships, e.g., Yingkou city has a spillover relationship to the air pollution of 35 other cities in the correlation network, while receiving urban pollution from 25 other cities. Yingkou city is affected by meteorological factors, such as the sea breeze due to its geographical location, which means it receives less pollution and exports more pollution compared to other cities. On the other hand, the cities of Changchun, Harbin, Siping, Shenyang and Liaoyuan are in an air pollution spillover position in the whole network due to their heavy industrial pollution. Daxinganling, Yichun, Shuangyashan, Hehe, Yanbian, Jiamusi, Hegang and Dandong are in the receiving position in the whole air pollution correlation network, such as Dandong City and 13 other cities for air pollution spillover relationship, while receiving air pollution spillover from 34 cities.

#### 3.3.3. Analysis of Cohesive Subgroups in Air-Polluted Cities in Northeast China

In this paper, based on the AQI index correlation data of 35 cities in Northeast China from January 2015 to June 2017, the Concor algorithm of Ucinet software is used to divide the 35 cities in Northeast China into four subgroups. As shown in Figure 3, Anshan, Changchun, Siping, Liaoyang, Chaoyang, Tieling, Liaoyuan, Yingkou, Jinzhou, Huludao andShenyang, are categorized in subgroup 1. These cities are located in the southwest of Northeast China, the border zone of Jilin-Liaoning and some cities in Liaoning Province, and the cities in subgroup 1 are mostly air pollution overflow roles in the correlation network, so subgroup 1 is an overflow subgroup. Songwon, Qiqihar, Harbin, Jilin, Daqing, Baicheng, Fuxin, Mudanjiang, Tonghua and Suihua are subgroup 2, which are mostly located in the central part of Northeast China, the border area of Jilin and Heilongjiang, and the cities in subgroup 2 are mostly “intermediaries” and “bridges” in the association network. Therefore, subgroup 2 is the “intermediary” subgroup. Benxi, Baishan, Heihe, Daxinganling, Jiamusi, Fushun, Yanbian, Hegang, and Dandong are sub-cluster 3, and Dalian, Jixi, Qitaihe, Yichun, and Shuangyashan are sub-cluster 4. Most of the cities in sub-cluster 3 and 4 are located in the northeast of Northeast China and are under the jurisdiction of Heilongjiang Province. These cities belong to the accepting role in the association network, so subgroups 3 and 4 are the accepting subgroups. The analysis results are more consistent with the previous point degree centrality analysis.

The 35 cities in Northeast China are divided into three categories: “spillover”, “intermediary”, and receiver of air pollution in the air pollution linkage network, and combined with the “role” cities. This paper attempts to analyze the transmission path of air pollution in the region by combining the geographical location of the “role” cities. The air pollution starts from the “spillover” subgroup cities (mostly in the southwest of Northeast China, mostly in Liaoning Province and the border cities of Jiliao), and travels through the “intermediary” subgroup cities (mostly in the central part of Northeast China and the border area of Jiliao and Heiliao) to the “receiving” subgroup cities (mostly in the central part of Northeast China and the border area of Jihe) to the “receiving” subgroup cities. In other words, from the analysis, the air pollution transmission route in Northeast China is from Liaoning region to Heilongjiang region via Jilin region. This is also consistent with the spatial clustering characteristics of air pollution in the region. Due to the natural geographic structure of Northeast China, which is surrounded by mountains on three sides and open in the middle, and the prevalence of northwesterly winds in autumn and winter when air pollution is severe. Therefore, the air pollution spreads from the “spillover” subgroup cities to the “intermediary” subgroup cities, and from the “intermediary” subgroup cities to the “receiving” subgroup cities. There is a tendency to shift to the southeast during the propagation process from the “spillover” subgroup cities to the “intermediary” subgroup cities and from the “intermediary” subgroup cities to the “receiving” subgroup cities. This also verifies that the point entry degree of Shuangyashan, Jiamusi and Jixi is larger than that of Daxinganling and other cities in the northwest direction in the previous analysis.

### 3.4. QAP Analysis of the Causes of Air Pollution Spatially Linked Pollutants in Northeast China

The purpose of QAP analysis is to investigate the regression relationship between multiple matrices and one matrix and to test the significance of the coefficient of determination R^2^. When both explanatory and explanatory variables are “relational variables” in the form of matrices, such relational data require independence among variables to avoid multicollinearity, which renders traditional econometric models and statistical tests useless. In contrast, QAP is an econometric method for analyzing relational data in social network analysis. It should be noted that the QAP regression model is a matrix regression of the relational data of the network, which is prone to problems, such as low R^2^ values.

Given that the air data, in this paper, are relational and in matrix mode, QAP regression analysis in social network analysis was chosen to analyze the influencing factors of spatial association of air pollution in Northeast China from the perspective of pollutants, and the regression results are shown in Table 3.

The regression coefficients of the spatial association of air pollution in Northeast China passed the 5% significance level test for all pollutants except SO_2_ and the coefficients were positive, among which PM_2.5_ and PM_10_ both passed the 1% significance level test. Among all the pollutants, the coefficient of PM2.5 is much higher than that of other pollutants at 0.462, indicating that PM_2.5_ is the main cause of spatial association of air pollution in the region, followed by PM_10_ at 0.273. It can be seen that these two pollutants dominate the spatial association of air pollution in the region. Among all pollutants, the magnitude of spatial association influence on air pollution is PM_2.5_ > PM_10_ > NO_2_ > O_3_ > CO > SO_2_. The adjusted R^2^ of QAP regression results is 0.347, which means that the model can explain approximately 34.7% of the spatial association causes of air pollution. Air pollution is generated by a combination of factors, including local geographical factors, such as temperature, humidity and altitude, as well as socio-economic factors such as pollution emissions and residential emissions. However, the study of spatial association of air pollution can effectively help to explore the control methods of regional pollution. For the results of QAP regression analysis, if the pollution relationship in the association network can be effectively controlled, it will be able to reduce about 34.7% of regional air pollution and greatly improve the air quality.

## 4. Conclusions and Recommendations

### 4.1. Conclusions

(1)Air pollution in Northeast China is generally correlated among cities in the region, showing a complex correlation network that transcends geographical distance, and the correlation network of air pollution has strong stability. The spatial correlation networks of PM_2.5_, PM_10_, and NO_2_ are significantly more stable than those of the other three pollutants among all sub-pollutants. The correlation degree of air pollution network is high, and the connectivity of the network is good. There is an obvious “small-world effect” in the correlation network, and only (1.1–1.4) intermediate cities are needed to establish the relationship between any two urban nodes in the region for all pollutants on average, and the cohesion index of all pollutants is between (0.82–0.93), which proves that the spatial correlation network has a very strong cohesiveness.(2)Different cities play different “roles” in the spatial association network of air pollution in Northeast China, and there are significant locational differences. The analysis of the cohesive subgroups shows that the 35 cities in the Northeast region can be roughly divided into the role of air pollution spillover, air pollution intermediary, and air pollution receiver. An attempt to analyze the air pollution transmission paths in the region shows that air pollution in Northeast China spreads from Liaoning region to Heilongjiang region through Jilin region.(3)Among the six pollutants, all except SO_2_ have a significant positive effect on AQI, and among all pollutants, the coefficient of PM_2.5_ is 0.462, which is much higher than the level of other pollutants, followed by PM_10_ at 0.273. It can be seen that these two pollutants dominate the spatial association of air pollution in the region. The QAP regression also shows that PM_2.5_ is the main cause of the spatial association of air pollution in cities. Among all pollutants, the magnitude of influence on the spatial association of air pollution was PM_2.5_ > PM_10_ > NO_2_ > O_3_ > CO > SO_2_.

### 4.2. Recommendation

(1)In response to the spatially linked network of urban air pollution existing in Northeast China, targeted joint air pollution prevention and control policies should be adopted, joint air pollution prevention and control prevention and control mechanisms should be developed, and large regional cooperation groups should be constructed to jointly combat air pollution problems. In the face of the complex correlation network of air pollution, any city’s air management actions alone will be offset by other cities’ pollution through the correlation network. Therefore, building a cross-regional joint air pollution prevention and control system is the fundamental solution to the air pollution problem. Based on the results of the analysis in this region, similar environmental protection measures should be taken in other regions such as Beijing, Tianjin and Hebei.(2)The role played by different cities in the air pollution linkage network is different. Therefore, when formulating cross-regional joint prevention and control policies, different air management policies should be developed for different cities to achieve fundamental control of air pollution problems in the Northeast. In the treatment of air pollution, a collaborative defense system focusing on PM_2.5_ and supplemented by PM_10_ and NO_2_ should be formed to effectively control the source of air pollution. In the Northeast region to develop cross-regional air pollution management policy, should focus on the Liaoning region air pollution “source” management, and in Jilin and Heilongjiang and other areas of air dispersion related means to intercept, for air pollution transmission path, effective interception management.(3)The management of air pollution is inseparable from the constraints of law and policy regulation. The effective introduction of air pollution prevention and control law is a necessary means of managing air pollution. Local governments should actively play a guiding and supervisory role in the process of air pollution management, targeted development of political and legal regulations in line with the specific characteristics of the local, strengthen environmental protection publicity, raise public awareness of environmental protection, promote green consumption, green lifestyle and mobilize the public to participate in the prevention and control of air pollution enthusiasm.

## Figures and Tables

**Figure 1 ijerph-18-10619-f001:**
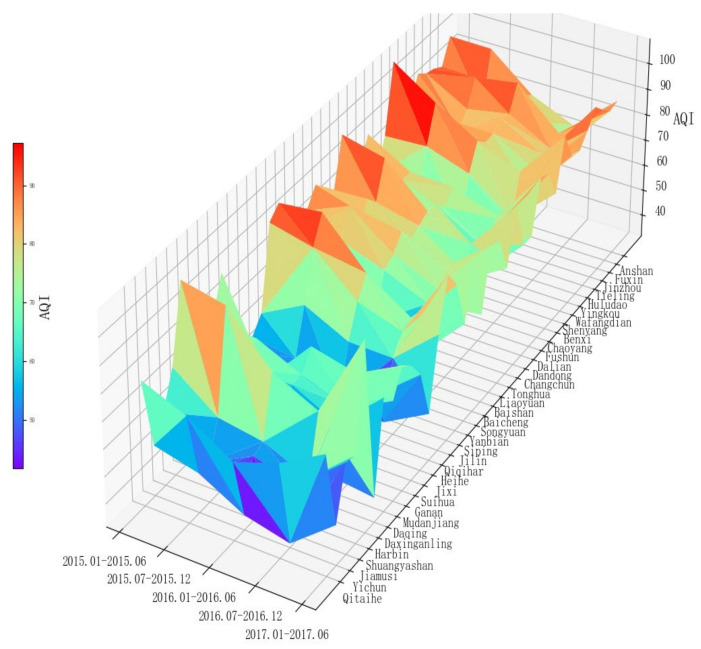
AQI Map of Northeast China’s Prefectural Cities.

**Figure 2 ijerph-18-10619-f002:**
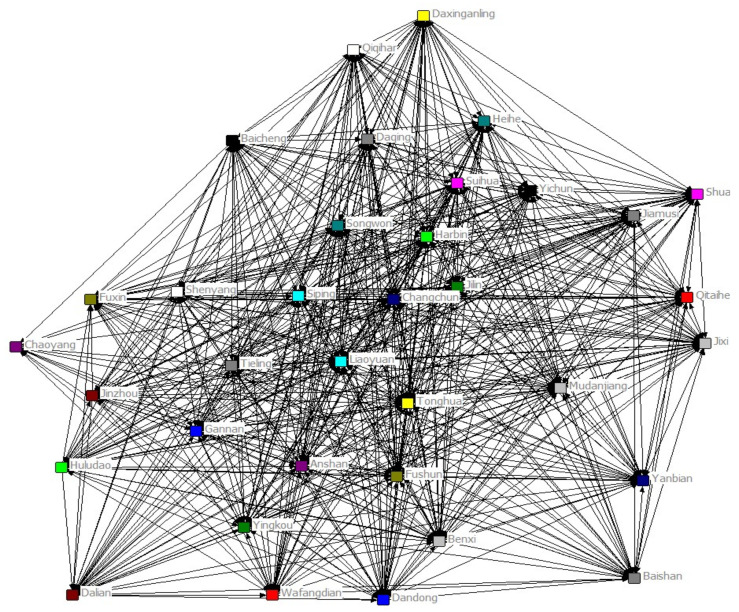
Spatial correlation network of urban air pollution in Northeast China.

**Figure 3 ijerph-18-10619-f003:**
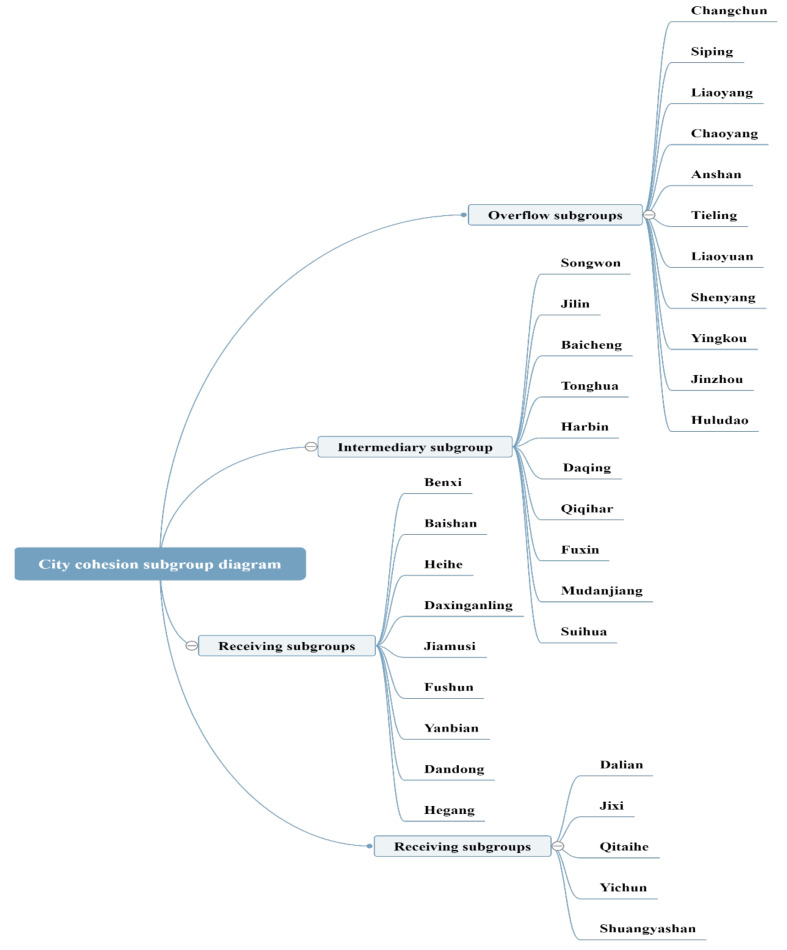
A cluster analysis of haze pollution network in northeast China.

**Table 1 ijerph-18-10619-t001:** The correlation network feature table of air pollution in northeast China.

	AQI	PM_2.5_	PM_10_	SO_2_	CO	NO_2_	O_3_
Network Density	0.769	0.826	0.758	0.764	0.735	0.726	0.623
Network Efficiency	0.682	0.719	0.752	0.775	0.762	0.757	0.825
Network Relevance	1	1	1	1	1	1	1
Average Distance	1.232	1.162	1.241	1.253	1.268	1.269	1.362
Cohesion Index	0.893	0.917	0.867	0.852	0.867	0.843	0.821

**Table 2 ijerph-18-10619-t002:** The characteristics of spatial association of haze pollution in northeast China (taking AQI as an example).

City Nodes	Point-Out Degree (Number of Overflow Associations)	Point-in Degree (Number of Accepted Associations)	Point Centrality	Intermediate Centrality
Yingkou	35	25	100	0.078
Changchun	35	21	100	0.078
Harbin	35	27	100	0.078
Siping	35	25	100	0.078
Shenyang	34	21	100	0.078
Liaoyuan	34	26	100	0.078
Suihua	34	24	100	0.078
Qiqihar	33	25	100	0.078
Qitaihe	33	33	100	0.078
Baicheng	33	27	100	0.078
Mudanjiang	32	32	100	0.078
Daqing	32	29	100	0.078
Tieling	32	18	91.429	0.022
Jilin	31	31	100	0.078
Chaoyang	30	14	91.429	0.045
Songwon	30	28	100	0.078
Liaoyang	30	27	94.286	0.033
Tonghua	30	29	100	0.078
Anshan	29	28	97.143	0.062
Jinzhou	28	17	91.429	0.021
Benxi	28	30	100	0.078
Fushun	25	28	97.143	0.049
Dalian	25	31	100	0.078
Huludao	24	20	85.714	0.016
Fuxin	24	22	94.286	0.039
Jixi	23	34	100	0.078
Baishan	23	32	100	0.078
Daxinganling	22	27	88.571	0.011
Yichun	20	31	100	0.078
Shuangyashan	20	35	100	0.078
Heihe	19	27	97.143	0.062
Yanbian	16	35	100	0.078
Jiamusi	16	35	100	0.078
Hegang	15	34	100	0.078
Dandong	13	34	100	0.078

**Table 3 ijerph-18-10619-t003:** QAP regression results of the spatial correlation of air pollution in northeast China.

Variable Matrix	Coefficient	*p*-Value
CO	0.0453	0.047
NO_2_	0.0617	0.011
O_3_	0.0513	0.028
PM_10_	0.273	0.006
PM_2.5_	0.462	0.000
SO_2_	−0.000057	0.503
Intercept	0.079662	0.391
R^2^	0.350	
Adj-R^2^	0.347	

## Data Availability

The raw/processed data required to reproduce these findings cannot be shared at this time as the data also forms part of an ongoing study.

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
