# Peer review of "Spatial Correlation of Air Pollution and Its Causes in Northeast China"

_ijerph, 2021, doi:10.3390/ijerph182010619_

Round 1

Reviewer 1 Report

Dear authors,

All comments, suggestions and considerations to the manuscript are in the file and marked with colors.

Reviewer 2 Report

The abstract should contain a defined research problem, research goal, indicate the research methodology, outline the results and conclusions to interest the potential reader. It is difficult to find these elements in the presented abstract.

I believe that what matters is the number of degrees of freedom (30765), not the data units you have.

In the sentence: According to the results in Table 2, the average distance of AQI in the northeast region is 1.232 (...) ”And further, the author refers to Table 2, in which this data cannot be found.

References to figures and tables are very distant - e.g. on page 10 you refer to Figure 3 which is on page 12.

It is difficult to conclude that the conclusion: PM2.5> PM10> NO2> O3> CO> SO2 is wrong, but we take into account the combination of positive and negative values. Is their interpretation the same?

I believe that the model - due to the clearly insignificant variable (SO2) should be estimated again without this variable to avoid its impact on other variables.

The author mentions other studies earlier, but did not discuss other results after presentation of the own.

Table 3 presents the results of the estimation, however, apart from R2, no test results were presented to verify the quality of the model.
